# Validation of the Spanish Version of Newcastle Stroke-Specific Quality of Life Measure (NEWSQOL)

**DOI:** 10.3390/ijerph17124237

**Published:** 2020-06-14

**Authors:** Concepción Soto-Vidal, Soraya Pacheco-da-Costa, Victoria Calvo-Fuente, Sara Fernández-Guinea, Carlos González-Alted, Tomás Gallego-Izquierdo

**Affiliations:** 1Department of Nursing and Physiotherapy, University of Alcala, 28871 Madrid, Spain; conchi.soto@uah.es (C.S.-V.); victoria.calvo@uah.es (V.C.-F.); tomas.gallego@uah.es (T.G.-I.); 2Department of Experimental Psychology, Complutense University of Madrid, 28040 Madrid, Spain; sguinea@psi.ucm.es; 3Medical Director State Reference Center for Brain Damage Care, 28034 Madrid, Spain; cgonzaalted@imserso.es

**Keywords:** quality of life, stroke, validation, psychometric properties

## Abstract

Background: Stroke causes a wide variety of clinical manifestations that may have a negative impact on quality of life. Therefore, it is very important to use specific instruments for measuring quality of life in individuals who suffered a stroke. The aim of this study was to develop a psychometrically validated Spanish version of the Newcastle stroke-specific quality of life measure (NEWSQOL). Methods: A psychometric validation of the Spanish version of the NEWSQOL questionnaire was carried out in 159 patients. The reliability (intraclass correlation coefficient and Cronbach’s alpha coefficient), validity (factorial analysis and Spearman’s coefficient), feasibility (response rate), and the ceiling and floor effects were calculated. Results: Internal consistency showed that Cronbach’s alpha coefficient was 0.93. The test–retest reliability was high or excellent for all domains (range 0.71–0.97 *p* < 0.001). The response rate of the questionnaire was 100% and the average administration time was 20.5 (±7.2) min. No ceiling effect was detected and two domains (pain and vision) may have a significant potential for floor effect. Construct validity showed that all the variables are important enough to keep them all in the questionnaire. Concerning convergent construct validity, a high correlation was found with the Nottingham Health Profile, the Barthel Index, and the Modified Rankin Scale. Conclusion: The Spanish version of the NEWSQOL questionnaire is reliable, valid, and feasible to evaluate quality of life in the Spanish population.

## 1. Background

The World Health Organization defines stroke as “the rapid onset of clinical symptoms of focal or global brain dysfunction, which lasts longer than 24 h or leads to death, with no other apparent cause than an injury vascular” [1]. Epidemiological studies situate the worldwide incidence of stroke in about 250 cases per 100,000 inhabitants/year [2]. It is the second cause of death and one of the main reasons of disability worldwide [3,4]. In Spain, about 4% of the people over 65 years of age live with very disabling sequelae of stroke [5,6]. There is a wide variety of clinical manifestations, such as motor, sensory, perception and cognitive disturbances, urinary and fecal incontinence, swallowing and visual problems, pain, communication, and behavioral alterations [7,8,9,10,11,12,13,14,15,16]. These symptoms affect activities of daily life (ADL), increase the risk of falls, and have a negative impact on quality of life (QoL) [17,18,19,20,21,22]. 

It is necessary to design and assess the effectiveness of appropriate and individual therapeutic interventions, in order to understand patients’ perceptions after suffering a stroke. Therefore, the aim of the interventions is not only to carry out clinical assessments, but also to evaluate the subjective patients´ perceptions. As Altal pointed out, “the effects of a treatment on Stroke should be measured in terms of QoL, in addition to disability and survival” [23]. This is the real importance of studying QoL perception in stroke.

Most studies about QoL in patients who had suffered a stroke were carried out with generic QoL instruments that were used to compare different populations and diseases [24]. The inclusion of more subjective aspects started to rise at the beginning of the century, but specific instruments remain scarce and many of them do not include all the dimensions that may be affected after suffering a stroke [25].

The original Newcastle stroke-specific quality of life measure (NEWSQOL) is a specific questionnaire to measure QoL in people who suffered a stroke. Buck et al. [26] developed NEWSQOL using patient-centered methods to encompass the symptoms after a stroke. In fact, NEWSQOL is different from others questionnaires, because it includes domains about communication, cognition, and vision, and it can be used for patients with ischemic or hemorrhagic stroke, and motor aphasia. The Spanish version of the NEWSQOL was culturally adapted for the Spanish population [27], using the original questionnaire as a reference.

Since there is a lack of specific tools for assessing QoL perception after a stroke that can collect information from the patient´s perspective [28], the aim of this paper was to develop a psychometrically validated Spanish version of NEWSQOL, in order to use it as a reference for assessing stroke patients´ QoL in the Spanish population.

## 2. Materials and Methods

A psychometric validation of the Spanish version of NEWSQOL questionnaire was carried out between July 2011 and February 2017. The present study was approved by the Research Ethics Committee at Ramon y Cajal University Hospital in Madrid, Spain. Participants were recruited by neurologists and neurological physical therapists from 5 different reference centers in Spain: 3 hospitals (Ramon y Cajal, Principe de Asturias, and Beata Maria Ana de Jesus); Brain Damage State Centre (CEADAC); and the Institute of Neurological Diseases (IEN). All subjects who had suffered a stroke between 1 month and 2 years at the time of recruitment were included in the study by non-probabilistic sampling. The exclusion criteria was: dependency for ADL before the stroke, suffering from stroke more than once, other neurological or neuromuscular condition, moderate or severe cognitive deterioration according to Pfeiffer’s questionnaire [29], psychiatric pathology previously diagnosed, neoplasia, or other serious illness with significant repercussions on QoL. After signing the written informed consent for participating in the study, all the data collected from each participant was associated with a code for anonymity guarantee. The questionnaires were administered through individualized interview with each patient.

Demographic data and clinical history of the patients was collected and those subjects who met the inclusion criteria answered the Spanish version of NEWSQOL questionnaire through an individualized interview.

NEWSQOL has 56 items in 11 domains: mobility (items 1–9); ADL/self-care (items 10–17); pain/sensation (items 18–20); vision (items 21–22); cognition (items 23–27); communication (items 28-31); feelings (items 32–37); interpersonal relationships (items 38-43); emotions (items 44–47); sleep (items 48–53); and fatigue (items 54–56). Each item is rated in range 0 to 3, and they are not significant individually. The results of each domain are obtained by the sum of the scores of the items, and higher values indicate greater impact on quality of life perception. The author of the original questionnaire does not recommend adding the scores obtained in the domains to achieve a global score [26].

Reliability was studied by test–retest reliability and internal consistency. Intraclass correlation coefficient (ICC) was performed for the test–retest reliability. It was calculated from the answers of the first 30 patients who came for a second interview, in a time interval between 7 and 12 days from the first interview, to answer to the questionnaire again, following the recommendations of Ramada-Rodilla et al., 2013, [30]. This time interval was selected in order to avoid complications that could arise in the follow-up of patients, such as hospital discharge, clinical problems, etc. Internal consistency was calculated by means of Cronbach’s alpha (α), in which values greater than 0.70 show good reliability [31].

Validity was assessed by convergent construct and factorial construct. Content validity was assessed in the original questionnaire and during the process of NEWSQOL cross-cultural adaptation to Spanish [27]. In order to evaluate convergent construct validity through the correlation with the Spanish version of NEWSQOL, all participants were also administered the Nottingham Health Profile (NHP), the Barthel Index (BI), and the Modified Rankin Scale (MRS). All of them are validated in the Spanish population [32,33,34]. NHP is a self-administered questionnaire designed to measure a patient’s view of their own health status. It consists of two parts, the first part with 38 statements belonging to six dimensions of health (energy, pain, emotional reactions, sleep, social isolation, and physical mobility), and the second contains seven areas related to ADL [32]. BI was developed to assess self-care and functional independence [33], and MRS is used for measuring the degree of disability or dependence in ADL in post-stroke patients [34,35]. The correlation between of the scores of the Spanish version of NEWSQOL with NHP, BI, and MRS was calculated. The statistic used was the Spearman correlation (r) for non-parametric tests. It was considered to have high validity when the correlation was between 0.30 and 0.40 [26]. For factorial construct validity, an analysis of the items was performed by the main components method, previously checking the adequacy of the sample to the factorial analysis, by means of the Bartlett Sphericity test and the measurement of the adequacy of Kaiser–Meyer–Olkin (KMO coefficient).

Feasibility was evaluated by percentage of individual items that were not answered, the percentage of patients who did not answer any items, and the average time required to fulfill the questionnaire. Ceiling and floor effects were analyzed to measure the percentage of patients with the best and worst possible score obtained. Ceiling or floor effects are considered when more than 20% of the responses get the best or worst possible score respectively [26].

IBM® (SPSS Statistics, Armonk, New York, USA) was used for the statistical analysis of the data obtained, a *p*-value of <0.05 was considered statistically significant for all evaluations.

## 3. Results

A total of 168 subjects were recruited for the study (3 per each questionnaire item); 9 did not accept to participate. Table 1 shows the socio-demographic and clinical characteristics of the 159 subjects who participated in this study. Most of the subjects were middle-aged (52 years old) men (56.6%) who were under sick leave (65.4%) and had help from their families (89.3%).

Results concerning reliability (test–retest reproducibility and internal consistency), and feasibility (floor/ceiling effects) are shown in Table 2. The Spanish version of the NEWSQOL questionnaire showed high internal consistency, with a Cronbach’s α coefficient of 0.93 (considering good reliability of values greater than 0.70), and a range between 0.57 and 0.96 for the domains. Intraclass correlation coefficient (ICC) showed good to excellent reproducibility test–retest for all domains (0.70–0.97), considering good values higher than 0.75. All values were statistically significant (*p* < 0.001). There were no ceiling effects, although floor effects were found for the domains pain, vision, interpersonal relationships, and fatigue. The rate of the fulfilling of the questionnaire was 100%, and the average time was 20.5 (±7.2) min.

Regarding the factorial construct validity, Table 3 shows the values obtained in the Bartlett sphericity test and the Kaiser–Meyer–Olkin adaptation measure (KMO). The analysis of the main components of the questionnaire indicated that 14 components explain 73.96% of the variance. This fact shows that all items contribute, at least, in one of the domains.

Table 4 shows high convergent construct validity values and all domains of the Spanish version of the NEWSQOL had a positive correlation with NHP and MRS scores, which shows that the higher the functionality, the higher the QoL. The higher Spearman coefficient of NHP was related to mobility domains, r = 0.688 (*p* < 0.01). The Spearman coefficient of MRS was high in mobility domain r = 0.871 (*p* < 0.01), ADL r = 0.770 (*p* < 0.01), and feelings r = 0.469 (*p* < 0.01). It was also moderate to pain domains r = 0.244 (*p* < 0.01), cognition r = 0.207 (*p* < 0.01), interpersonal relations r = 0.305 (*p* < 0.01), communication r = 0.240 (*p* < 0.01), vision r = 0.186 (*p* < 0.01), and fatigue r = 0.218 (*p* < 0.01).

The Spanish version of the NEWSQOL had a negative correlation with the BI. Spearman coefficient was high for the domains of mobility r = −0.883 (*p* < 0.01), limitation for ALD r = −0.808 (*p* < 0.01), and it was moderate for domains of feelings r = −0.382 (*p* < 0.01), interpersonal relations r = −0.251 (*p* < 0.01), communication r = −0.277 (*p* < 0.01), cognition r =-0.184 (*p* < 0.01), and vision r = −0.197 (*p* < 0.01). The Spearman coefficient was low to pain r = −0.146 (*p* < 0.05) and fatigue r = −0.151 (*p* < 0.05) (Table 5).

## 4. Discussion

As far as the authors know, this is the first validation study for the NEWSQOL questionnaire in another language than English, since no other cross-culturally adaptations nor validations were found in the literature.

In the recent years, various countries have been using this instrument for measuring quality of life [10,20]. The perception of the patients´ perspective about their illness allows the knowledge of the clinical decision on the best intervention for patients with a chronic disease, in order to improve their quality of life [36,37]. The NEWSQOL is a recommended measure to assess QoL for many reasons. First of all, because it was developed following patient-centered methods and it includes all the relevant aspects, from patients´ point of view and perspective, which may be present after a stroke. Besides, it can be administered to patients who suffered either hemorrhagic or ischemic stroke, since the clinical manifestations are similar [7,8,9,19]. The present study followed the original questionnaire methodology.

The lack of cultural adaptation of the instrument in other countries made it the impossible to use other studies’ sample size as reference. In the literature, the studies about psychometric validation of specific questionnaires for stroke showed a wide variability of sample size [38,39,40,41,42,43]. In the present study, the authors decided to recruit three patients for each item of the original questionnaire, following Argimon’s methodology [44], which states that a questionnaire has to have a sample of between 2 and 10 subjects per questionnaire item. The sample size was larger than other validation studies of specific questionnaires for stroke [45] and it is similar to other studies of patients with stroke, regarding age, type of stroke, hemibody affected, comorbidity, and family situation [2,5,7,8,11,12,13,14,15,21,46]. The patients were recruited from different reference centers that receive patients from all over Spain, because they have the most innovative treatments to address stroke, which could ensure a wider geographical distribution of the sample.

Unlike other specific instruments for measuring QoL after a stroke, the NEWSQoL includes the dimensions of communication, cognition, and vision, which are relevant aspects for stroke survivors. The instrument can also be used for both, ischemic and hemorrhagic strokes, and for patients with motor aphasia.

The results found in the study were very satisfactory. As for viability, the high level of response (100%) may have been influenced by administering the questionnaire through personal and individualized interviews. The data of effect ceiling and ground coincide with those provided by the original questionnaire [26] showing that there is no ceiling effect in any domain, but floor effects were found for the domains pain, vision, interpersonal relations, and fatigue. The time for its completion also coincide.

Regarding reliability, the questionnaire as a whole showed excellent results on internal consistency. When the domains were analyzed independently, there were also excellent or very good correlations, except for the domains of emotions and pain, which showed moderate internal consistency. Test–retest reliability showed good or excellent correlation for all domains. These results coincide with those provided by the original questionnaire and the Spanish cultural adaptation study [27].

Construct validation highlighted that all variables have enough importance, so that they all must be in the questionnaire, due to their contribution to one of the 11 dimensions. Factorial analysis supported its one-dimensionality, as in the original questionnaire [26].

The results of the convergent construct validity indicated high correlation between the domains that measure the same aspects. It seems logical that a mobility deficit involves greater difficulty in performing ADL and may have an impact on sleeping, social isolation, and emotional reactions, but it does not necessarily involve increased pain [13]. The pain correlated with physical mobility, and may affect sleep, energy, and emotional reactions. The domain of vision showed moderate correlation with physical mobility and social isolation, which may be reduced by visual deficit [19,47]. Memory and concentration showed correlation with physical mobility, pain, energy, physical isolation, and emotional reactions [8,9]. Communication had a correlation with social isolation [16,48], and feelings showed correlation with social isolation and emotional reactions [49]. Fatigue showed correlation with energy, emotional reactions, physical mobility, and social isolation similar to the original questionnaire [26].

Correlations with MRS emphasize the idea of not associating disability only to physical limitation, but also to social activity, interpersonal relations, cognition, vision, and communication, aspects that may lead to the perception of disability [50]. Regarding the BI, there was no correlation with sleep and emotion.

## 5. Conclusions

The Spanish version of the NEWSQOL questionnaire is reliable, valid, and feasible for assessing QoL after a stroke in the Spanish population. Consequently, this questionnaire may be useful in Spanish-speaking populations and for making cross-ethnic and cross-cultural comparisons with other English-speaking countries that have a large Spanish-speaking population. Besides, as this is the first validation study for the NEWSQOL in another language, it may be a reference frame for other cross-culturally adaptations and validations in other countries.

## Figures and Tables

**Table 1 ijerph-17-04237-t001:** Socio-demographics and clinical characteristics of the sample.

Baseline Characteristics of the Sample	
**Sex (%)**	
Women	43.4
Men	56.6
Age [Mean (SD)]	52.09 (±15.72)
**Type of Stroke (%)**	
Ischemic	55.3
Hemorrhagic	44.7
**Hemibody affected (%)**	
Left	47.2
Right	45.3
Both	7.5
**Time** **after stroke** **[Mean (SD)]**	7.58 (±5.72)
**Employment** **Situation (%)**	
Active	0
Unemployed	7.5
Retired	26.4
Sick leave	65.4
**Family** **situation**	
Family help	89.3
Lives alone	1.3
Institutionalized	8.8

**Table 2 ijerph-17-04237-t002:** Values of internal consistency, test–retest reliability, ceiling, and floor effect.

NEWSQOL Dimensions	Internal Consistency	Test–Retest	Ceiling/Floor Effect
	(*n* = 159)	(*n* = 30)	
	Cronbach’s α	ICC	(%)
NEWSQOL	0.94		
Mobility	0.96	0.96	16/3
ADL/self-care	0.92	0.97	22/3
Pain /sensation	0.68	0.70	2/62
Vision	0.79	0.83	5/45
Cognition	0.86	0.84	6/9
Communication	0.88	0.95	18/13
Feelings	0.78	0.87	4/0
Interpersonal Relationship	0.79	0.88	0/35
Emotion	0.57	0.73	2/0
Sleep	0.78	0.73	1/3
Fatigue	0.66	0.80	6/38

ICC: intraclass correlation coefficient; ADL: activities of the daily life.

**Table 3 ijerph-17-04237-t003:** Values of factorial construct validity of the Newcastle stroke-specific quality of life measure (NEWSQOL).

NEWSQOL Dimensions	Bartlett	KMO	Number of Items	Total Variance %
NEWSQOL	0.000	0.809	56	73.96
Mobility	0.000	0.933	9	70.98
ADL/self-care	0.000	0.885	8	79.20
Pain /sensation	0.000	0.508	3	61.32
Vision	0.000	0.500	2	81.38
Cognition	0.000	0.686	5	63.65
Communication	0.000	0.801	4	72.97
Feelings	0.000	0.712	8	56.80
Interpersonal relationships	0.000	0.766	6	52.32
Emotion	0.000	0.607	4	43.92
Sleep	0.000	0.698	6	85.20
Fatigue	0.000	0.559	3	57.52

KMO: Kaiser–Meyer–Olkin; ADL: activities of daily life.

**Table 4 ijerph-17-04237-t004:** Values of convergent construct validity (Spearman’s coefficient) of the NEWSQOL, Nottingham Health Profile, and Modified Rankin Scale.

NEWSQOL	NHP						EMR
	PM	Pain	Sleep	Energy	ER	SI	
Mobility	0.688 **	0.098	0.308 **	0.192 **	0.304 **	0.343 **	0.871 **
ADL/self-care	0.615 **	0.104	0.233 **	0.179 *	0.336 **	0.280 **	0.770 **
Pain /sensation	0.337 **	0.722 **	0.355 **	0.280 **	0.136 *	0.283 **	0.244 **
Vision	0.297 **	0.093	0.123	0.104	0.198 **	0.150 *	0.186 **
Cognition	0.262**	0.218 **	0.044	0.228 **	0.276 **	0.319 **	0.207 **
Communication	0.166 *	−0.028	−0.014	−0.004	0.288 **	0.113	0.240 **
Feelings	0.311 **	0.217 *	0.311 **	0.343 **	0.470 **	0.615 **	0.469 **
Interpersonal relationships	0.333 **	0.200 **	0.246 **	0.313 **	0.515 **	0.644 **	0.305 **
Emotion	0.239	0.066	0.206	0.497 **	0.494 **	0.484 **	0.255
Sleep	0.239	0.066	0.206	0.497 **	0.494 **	0.484 **	−0.041
Fatigue	0.272 **	0.162 *	0.083	0.535 **	0.256 **	0.422 **	0.218 **

NHP: Nottingham Health Profile; MRS: Modified Rankin Scale; PM: Physical mobility; ER: Emotional Reactions; SI: Social Isolation; ADL: activities of daily life; * *p* ≤ 0.05; ** *p* ≤ 0.01.

**Table 5 ijerph-17-04237-t005:** Values of Convergent construct analysis (Spearman’s coefficient) of the NEWSQOL and Barthel Index.

Newsqol	Barthel Index	Feeding	Bathing	Grooming	Dressing	Boewls	Bladder	Toilt Use	Transfers	Mobility	Stairs
Mobility	−0.883 **	−0.520 **	−0.726 **	−0.774 **	−0.414 **	−0.377 **	−0.301 **	−0.786 **	−0.839 **	−0.868 **	−0.871 **
ADL/self-care	−0.808 **	−0.566 **	−0.679 **	−0.802 **	−0.367 **	−0.254 **	−0.199 **	−0.711 **	−0.777 **	−0.755 **	−0.747 **
Pain /sensation	−0.146 *	−0.161 *	−0.077	−0.079	−0.212 **	−0.198 **	−0.086	−0.085	−0.103	−0.138 *	−0.097
Vision	−0.197 **	0.000	−0.091	−0.170 *	−0.305 **	−0.246 **	−0.124	−0.204 **	−0.228 **	−0.191 **	−0.197 **
Cognition	−0.184 *	−0.123	−0.094	−0.112	−0.145 *	−0.210 **	−0.187 **	−0.156 *	−0.164 *	−0.153 *	−0.164 *
Communication	−0.277 **	−0.154 *	−0.159 *	−0.170 *	−0.310 **	−0.204 **	−0.166 *	−0.199 **	−0.306 **	−0.212 **	−0.265 **
Feelings	−0.382 **	−0.224 **	−0.380 **	−0.387 **	−0.163 *	−0.095	−0.152 *	−0.343 **	−0.392 **	−0.311 **	−0.369 **
IR	−0.251 **	−0.051	−0.300 **	−0.216 **	−0.137 *	−0.011	−0.078	−0.146 *	−0.216 **	−0.216 **	−0.279 **
Emotion	−0.322 *	−0.162	−0.256	−0.459 **	−0.084	0.029	−0.060	−0.342 *	−0.381 *	−0.304	−0.155
Sleep	−0.170	−0.151	−0.033	−0.128	−0.154	−0.264	−0.503 **	−0.135	−0.111	−0.043	−0.050
Fatigue	−0.151	−0.057	−0.107	−0.179 *	−0.099	−0.133 *	−0.134 *	−0.210 **	−0.141 *	−0.116	−0.151 *

ADL: activities of daily life; IR: Interpersonal relationships; * *p* ≤ 0.05; ** *p* ≤ 0.01.

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
