# Peer review of "Validation of the Spanish Version of Newcastle Stroke-Specific Quality of Life Measure (NEWSQOL)"

_ijerph, 2020, doi:10.3390/ijerph17124237_

Round 1

Reviewer 1 Report

Thank you for inviting me to review this study.

Manuscript ID ijerph-823070 entitled “Validation of the Spanish version of Newcastle 2 Stroke-specific quality of life measure (NEWSQOL)” by Soto-Vidal et al. have reported
a psychometric validation of the Spanish version of the NEWSQOL questionnaire, which has 56 items in 11 domains. The authors evaluated reliability, validity, and feasibility parameters of the questionnaire which was applied to patients who had Ischemic or hemorrhagic stroke. However, there are some points to be considered.

Abstract:

The structure required is Background, Methods, Results, and Conclusion.
Line 10  There is a semicolon after the ABSTRACT name.
The abstract should show the main results of the study.

Introduction:

The rationale is very logical and clear. It requires only one alteration. The authors informed the domains of the NEWSQOL questionnaire in the introduction (L48-51) and this was repeated in the Material and methods section (L72-75). This should only be demonstrated in the material and methods section.

Material and methods:

To evaluate the test-retest reliability, the authors only applied the questionnaire to the first 30 patients (18.86%) of the total sample (159 participants), which is a small sample to questionnaire validation study. The authors should explain what important aspects were considered to take this decision?

Another question that can be asked is which statistical analysis was performed to confirm the test-retest reliability?

Also, to clarify if the application between questionnaire 1 and questionnaire 2 was too short, the authors should demonstrate the difference of the time interval to each participant, because the duration between time 1 and time 2 was between 2 and 12 days. For example, How many participants had a two-day interval time to answer the questionnaire again?

L109. What was the SPSS version used in this study?

Results:

The table format should be uniform in the whole results section.

Despite the Barthel Index appears in Table 1, the data were not shown.

Cronbach’s α coefficient of 0.572 (L122) is not reported in Table 2.

Pay attention to the footnote of Table 2. The expression “activities of daily life” was written in gray.

Discussion:

L157 - Some word was missed. “The studies the were reviewed about….”
L160;L162 - There are parentheses in the presentation of references.

L165-168: The authors suggest that the Spanish- NEWSQOL questionnaire can be used for ischemic and hemorrhagic strokes. However, in this study, they did not compare with these groups or did not perform statistical analysis to show this evidence.

As general note, the study is easy to understand by the reader.

Author Response

Thank you for reviewing our paper - Manuscript ID ijerph-823070 entitled “Validation of the Spanish version of Newcastle 2 Stroke-specific quality of life measure (NEWSQOL)” by Soto-Vidal et al.

Reviewer 2 Report

Thank you for this work of high importance. My recommendations for improving the manuscript are:

  1. Kindly validate the study design: Is it a psychometric validation of a questionnaire, or the development (construction) of a questionnaire, or an observational transversal study?
  2. How was the original questionnaire translated? Was the new version validated by a group of experts?
  3. The sample size calculation needs to described and justified even for psychometric validation of a questionnaire. Did the author aim for 2 questionnaires or more (exact number) per questionnaire item?
  4. How were participants recruited? What are “different reference centers in Spain”? is it general practioners’ clinic (Setting)? Who recruited the participants?
  5. What were the inclusion criteria: individuals able to read and write in Spanish? Age group? Etc..
  6. How was data collected: was the questionnaire self-administered?
  7. Data management: How did the authors handle incomplete questionnaires? Anonymization? Etc
  8. As stated, in the manuscript, the author of the original questionnaire did not pursue a global score. Why was a global score calculated and reported for the Spanish version? (What are the authors’ arguments?)
  9. The presentations of the results' tables must be improved and can be better structured and have the same format.

Author Response

(The authors gave the same response as above.)

Reviewer 3 Report

Dear Author,

The manuscript is interesting, but it supports the previous article.

Soto-Vidal C, Pacheco-da-Costa S, Fernandez-Guinea S, Gallego-Izquierdo T. Translation into Spanish and a preliminary analysis of the psychometric properties of the Newcastle Stroke-Specific Quality of Life Measure (NEWSQOL) questionnaire for rating the quality of life among post-stroke patients. Rev Neurol 2017;65(11):481-488.

I suggest a little revision of English to remove some gaps, for example in line 13 (design).

As well as reading for the correction of some formatting mistakes, for example in line 123-126 (0,706 - incorrect / 0.706 – correct).

Would it be possible to explain this sentence better?

“The analysis by principal components of the questionnaire, 14 components explain the 73.96 of the variance.” (line 129-130)

I suggest a more careful presentation of table 5.

I suggest the revision of References formatting:

  • withdraw the month of publication (reference 10 (line 221) and 11 (line 224)
  • add publication date (reference 5 (line 210), 46 (line 305), and 47 (line 308).

The conclusion should be enriched.

Best regards.

Author Response

(The authors gave the same response as above.)

Round 2

Reviewer 1 Report

The authors improved their manuscript. In addition, this study should open the gate for a whole new line of research in their country.